# Construction of a *de novo* assembly pipeline using multiple transcriptome data sets from *Cypripedium macranthos* (Orchidaceae)

**Kota Kambara[1,2], Kaien Fujino[1], Hanako Shimura [1]***

**1** Faculty of Agriculture, Hokkaido University, Kita-ku, Sapporo, Japan, **2** Asian Natural Environmental Science Center (ANESC), The University of Tokyo, Nishitokyo, Japan

* hana@agr.hokudai.ac.jp

**Data Availability Statement:** All relevant data are within the paper and its Supporting Information files.

## Abstract

The family Orchidaceae comprises the most species of any monocotyledonous family and has interesting characteristics such as seed germination induced by mycorrhizal fungi and flower morphology that co-adapted with pollinators. In orchid species, genomes have been decoded for only a few horticultural species, and there is little genetic information available. Generally, for species lacking sequenced genomes, gene sequences are predicted by *de novo* assembly of transcriptome data. Here, we devised a *de novo* assembly pipeline for transcriptome data from the wild orchid *Cypripedium* (lady slipper orchid) in Japan by mixing multiple data sets and integrating assemblies to create a more complete and less redundant contig set. Among the assemblies generated by combining various assemblers, Trinity and IDBA-Tran yielded good assembly with higher mapping rates and percentages of BLAST hit contigs and complete BUSCO. Using this contig set as a reference, we analyzed differential gene expression between protocorms grown aseptically or with mycorrhizal fungi to detect gene expressions required for mycorrhizal interaction. A pipeline proposed in this study can construct a highly reliable contig set with little redundancy even when multiple transcriptome data are mixed, and can provide a reference that is adaptable to DEG analysis and other downstream analysis in RNA-seq.

## Introduction

A characteristic feature of the association between orchids and orchid mycorrhizal fungi (OM fungi) is the fungus-induced germination of orchid seeds, and thus the interaction with OM fungi is indispensable for the orchid life cycle [1–3]. Orchid seeds consist of a spherical, immature embryo with a seed coat; they lack storage tissues such as the endosperm and cotyledons normally found in seed plants [4]. In the process of germination, the embryo swells in size and develops into a protocorm before becoming a plantlet. Protocorm is a unique structure designed to establish interaction with OM fungi [5]. The protocorm lacks chlorophyll and is heterotrophic, so it receives nutrients from OM hyphae until shoots develop and photosynthesis begins. Although trehalose has been suggested as one of the major carbon sources supplied by OM fungi [6–9], details on the mechanism involved in nutrient acquisition remain unknown.

**Funding:** HS was supported by JSPS KAKENHI Grant Number JP17K19253. The funders had no role in study design, data collection and analysis, decision to publish, or preparation of the manuscript.

**Competing interests:** The authors have declared that no competing interests exist.

In more than 70% of land plants, roots are infected with arbuscular mycorrhiza (AM) fungi; in a mutualistic exchange, the AM fungi provide mineral nutrients to the plants, which supply carbohydrates to the fungi. In contrast, the relationship between orchids and OM fungi is not definitively mutualistic; some reports suggest the relationship is antagonistic rather than mutualistic, but others report that the interaction is similar to an AM interaction [10,11]. The molecular mechanisms involved in the interactions between some orchids and OM fungi have been studied using genomic and transcriptomic sequence data [9,11–14]. As for research on nutrient transfer, Fochi et al. (2017) analyzed the transcriptomes of *Tulasnella calospora*, a representative species of OM fungi, and suggested that the *T. calospora*-infected protocorms of *Serapias vomeracea* received organic N from the fungi [13]. Li et al. (2022) constructed genomes of two orchid species, partially mycoheterotrophic *Platanthera zijinensis* and holo-mycoheterotrophic *P. guangdongensis*, and showed the involvement of trehalase genes in mycoheterotrophy [9]. Regarding the regulation of mycorrhizal interactions with orchids, Perotto et al. (2014) analyzed the transcriptomes of *T. calospora*-infected protocorms of *S. vomeracea* and showed that some nodulin-like genes were upregulated in the fungus-infected region of protocorms [11]. Common symbiotic genes (CSGs) are known to have major roles in the establishment of rhizobium–legume symbiosis and AM symbiosis [15–18]. Miura et al. (2018) compared the expression patterns of different life stages of *Bletilla striata* protocorms infected by *Tulasnella* sp. and suggested that the CSGs are also involved in OM symbiosis [14]. On the other hand, in *Cymbidium hybridum*, transcriptome analysis suggested that mycorrhizal fungal infection induced the general defense responses in the orchid roots similar to infection by non-mycorrhizal fungi [19]. It has been shown that OM fungi species retain plant-cell-wall-degrading enzyme genes to the same extent as plant pathogenic fungi, while fungal species involved in ectomycorrhizal interactions have lost such genes during the co-adaptation of mycorrhizal interaction with host plants [20]. The relationship between orchids and fungi remains largely unexplored, and more research using diverse orchids and fungi is necessary to better understand the interactions between orchids and OM fungi.

As described above, the genomes have not been sequenced for most orchid species. For such non-model plant species, genomic information is constructed through the *de novo* assembly of transcriptome data. Because a universal assembly method for all organisms has not yet been established, various methods have been proposed for different species. Representative tools include Trinity [21], Trans-ABySS [22], rnaSPAdes [23], SOAP denovo-Trans [24] and Velvet [25], which use de Bruijn graphs for *de novo* assembly [26]. Each of the many assemblers that have been developed has its own advantages and disadvantages; so far, no tool can create a complete assembly [27]. Therefore, even with the same input data, the results will differ depending on the assembler used, and the results with one assembler will also differ depending on the selected settings for the parameters (e.g., k-mer), so reproducibility remains a problem [28,29]. In addition, output contigs obtained from *de novo* assembly are likely to contain redundancies, and such high-redundant data can be a major obstacle for downstream study [30].

For better assembly from transcriptome data, comparison of the efficacy of the tools, elimination of redundancy, and combining multiple tools have been proposed [27,30]. In the present study on *Cypripedium macranthos* var. *rebunense*, we analyzed the transcriptome of protocorms when seeds were (1) germinated aseptically, (2) germinated with an OM fungus but growth stopped, or (3) germinated with an OM fungus and growth continued, to search for genes important for the establishment and maintenance of mycorrhizal interaction. Because the genomic sequence is not available for *C. macranthos* var. *rebunense* or any close relatives, we *de novo* assembled and reconstructed genomic information from transcriptome

data. By integrating several assemblies obtained using different *de novo* assembly tools, we were able to construct a more complete and less redundant *de novo* transcriptome assembly.

## Materials and methods

### Plant material, RNA extraction, and sequencing

*Cypripedium macranthos* var. *rebunense* is maintained in the Botanic Garden, Hokkaido University. Mature seeds of *C. macranthos* var. *rebunense* were surface-sterilized essentially as described previously and sown on OMA2 medium for symbiotic germination with an OM fungus or on modified T-medium without yeast extract for aseptic germination [31,32] in a plastic culture plate. After cold treatment at 4°C in the dark for 9 weeks, the plates were transferred to 20°C in the dark and then incubated for 14 weeks to obtain protocorms. For symbiotic germination, OM fungal strain WO97 or FT061 were used; both have been confirmed to induce germination of *C. macranthos* var. *rebunense* [33]. WO97 and FT061 have been isolated from roots and germinated protocorms of *C. macranthos* var. *rebunense*, respectively [33], and assumed to be classified to an unidentified species in the genus *Tulasnella*. An agar block of a culture of either WO97 or FT061 was placed on seeds sown on an OMA2 plate after cold treatment. From each of the three treatments, 50–100 protocorms were collected for RNA extraction.

Total RNA was extracted and treated with DNase I as described previously [32], then the purified RNA was subjected to RNA-seq. Library preparation using the TruSeq Stranded mRNA Sample Prep Kit (Illumina, San Diego, CA) and sequencing using the Illumina HiSeq 2000 platform were outsourced to Hokkaido System Science Co., Ltd (Sapporo, Japan). The three libraries were sequenced yielding about 40 million 100-bp paired reads per library. Obtained raw reads were submitted to DDBJ Sequence Read Archive (DRA) as accession number DRA015853.

### Pre-processing sequence data and *de novo* transcriptome assembly

Raw reads were pre-processed using Trimmomatic v3.3 [34] and the Cutadapt program [35] to remove adapter sequence and low-quality reads, then processed using Rcorrector v1.0.5 [36] for error correction. For the *de novo* assembly, five transcriptome assemblers were used: Trinity v2.13.2 [21], rnaSPAdes v3.15.4 [23], Velvet v1.2.10 [25], Trans-ABySS v2.0.1 [22], IDBA-Tran v1.1.3 [37]. For all these assembly tools but Trinity, the k-mer value is set arbitrarily. Because k-mer length strongly affects the *de novo* transcriptome assembly [28], to avoid any effect of k-mer length, we constructed eight assemblies for each assembly tool (from k-mer of 25 to 95 with 10 steps) and then combined them. To combine assemblies obtained using the different k-mer lengths, we used CD-HIT-EST v4.8.1 (parameter -c is set to 0.99, according to [38]) for assemblies from rnaSPAdes and IDBA-Tran, and Oases v0.2.09 [29] for assemblies from Velvet. For assemblies from Trans-ABySS, we used the "merge" option included in the program. Because the k-mer length cannot be changed in Trinity (set at 25 k-mer), we did not use this combination process when using Trinity for assembly. Five assemblies obtained from each assembler were then merged into one assembly with 26 patterns using the EvidentialGene tr2aacds pipeline [39]; (http://arthropods.eugenes.org/EvidentialGene/about/EvidentialGene_trassembly_pipe.html).

### Evaluation of assemblies and gene annotation

The resulting assemblies were evaluated by contig number, mapping rate, and completeness. To assess completeness, we used Benchmarking Universal Single-Copy Orthologs (BUSCO)

v5.4.2 [40] and a BLAST search. BUSCO searches for universal single-copy orthologs in an assembly and estimates the completeness and redundancy of the assemblies. "Embryophyte" (land plants) of the OrthoDB v10 was used as a reference. For the BLAST search, we used DIA-MOND v2.0.13 [41] to search against the RefSeq plant database [42]. The E-value was set to 1e-20. Mapping rate is defined as percentage of reads aligned (mapped) to the constructed contigs among all reads, and was calculated using Salmon v1.7.0 [43]. The coding sequence (CDS) regions of the best-evaluated assembly were predicted by TransDecoder v 5.5.0 (https://github.com/TransDecoder/TransDecoder) using the results from the DIAMOND blastp search (E-value cutoff is 1e-20) against the RefSeq plant data and hammer v3.3.2 [44]. The contig set was then annotated against the RefSeq plant database using DIAMOND blastx and InterProScan v1.8.0 [45]. The contig set was submitted to DDBJ Transcriptome Shotgun Assembly (TSA) database as accession numbers ICTD01000001-ICTD01073457 (https://ddbj.nig.ac.jp/public/ddbj_database/tsa/TSA_ORGANISM_LIST.html). Annotation of the contigs from the DIAMOND blastx search was listed in S1 Table.

## Results

Observations of protocorm development of *C. macranthos* var. *rebunense* after inoculation with the OM fungi showed that growth differed depending on the fungal isolate. Representative images of protocorms are shown in S1 Fig; FT061-infected protocorms grew larger, while the WO97-infected protocorms appeared to stop growing and roots did not differentiate. We also prepared protocorms induced by aseptic germination for comparison, then used them for RNA extraction and subsequent RNA-seq.

For making an assembly, we assumed transcriptome data from multiple experimental conditions should be mixed to create a contig set that covers all genes encoded in the genome because genes that are expressed under certain experimental conditions may not be expressed under other conditions. We therefore carried out the *de novo* assembly with or without mixing raw reads. The total number of contigs obtained by *de novo* assembly without mixing raw reads is shown in Table 1. The five assembly programs generated more contigs from transcriptome data from the OM fungus-infected protocorms (Data sets 1 and 2 in Table 1) than from the aseptically grown protocorms (Data set 3). Among the five assemblers, Trinity yielded the fewest contigs (104,892–148,197) and Trans-ABySS the most (291,715–402,576). When three transcriptome data sets were mixed and then assembled, the number of total contigs increased in all cases (Table 1). Assessment by BUSCO showed that the rate of complete BUSCO (single-copy + duplicated) ranged from 81.0% (Velvet, WO97-infected protocorms) to 87.2% (rnaS-PAdes, aseptically grown protocorms) when a single data was used for assembly (Fig 1A), suggesting that the assembly seems incomplete because the percentage of the core gene was below 90%. However, by mixing three data sets from different conditions, the percentage of core genes in the contigs increased, exceeding 90%, except when using Velvet (89.7%) (Fig 1B). In the BLASTX search against the RefSeq plant database to estimate the number of genes in the contigs, the rate of the BLAST hits for the contigs obtained from a single data set was 19.8–45.2%, but 19.4–32.4% for contigs obtained from mixed reads (Table 1), which may be due to the increase in the total number of contigs. These results indicate that merging raw reads contributed to the completeness of the assembly, but increased the redundancy in the assembly.

To reduce the redundancy in the assembly without compromising completeness, we constructed a new pipeline for the *de novo* assembly of multiple transcriptome data sets. We used the EvidentialGene tr2aacds method to combine outputs from different assemblers and prepared 26 patterns of integrating assembly (Table 2; Fig 2). All the raw reads were mixed before the pre-processing step. In the 26 patterns (Assembly 1–26), the total number of contigs,

**Table 1. Evaluation of the assembly constructed by different assemblers using one or multiple transcriptome data sets.**

| Data set* | Assembler | Total no. of contigs | Mapping rate | N50 | % of BLAST hit contigs |
|---|---|---|---|---|---|
| 1 | TR | 1,48,197 | 96.3 | 1,557 | 37.3 |
| | RS | 2,36,340 | 97.0 | 1,024 | 23.6 |
| | VL | 2,65,676 | 68.3 | 1,053 | 34.2 |
| | TA | 3,56,100 | 89.1 | 1,044 | 21.7 |
| | IT | 2,09,149 | 96.6 | 1,428 | 37.5 |
| 2 | TR | 1,38,870 | 96.5 | 1,471 | 38.6 |
| | RS | 2,21,421 | 97.3 | 1,076 | 24.7 |
| | VL | 2,14,649 | 94.3 | 1,099 | 39.4 |
| | TA | 4,02,576 | 87.3 | 986 | 19.8 |
| | IT | 1,79,591 | 96.8 | 1,486 | 41.2 |
| 3 | TR | 1,04,892 | 96.9 | 1,385 | 42.7 |
| | RS | 1,43,823 | 97.4 | 1,051 | 30.2 |
| | VL | 1,66,189 | 66.8 | 1,044 | 43.5 |
| | TA | 2,91,715 | 90.8 | 908 | 22.3 |
| | IT | 1,42,133 | 97.3 | 1,420 | 45.2 |
| 1–3 combined | TR | 2,66,805 | 97.2 | 1,456 | 31.2 |
| | RS | 6,36,515 | 98.8 | 1,296 | 26.1 |
| | VL | 5,62,443 | 94.2 | 909 | 26.3 |
| | TA | 7,26,521 | 93.1 | 935 | 19.4 |
| | IT | 3,46,916 | 97.7 | 1,418 | 32.4 |

*Data set: 1, from WO97-infected protocorms; 2, from FT061-infected protocorms; 3, from aseptically grown protocorms. TR, Trinity; RS, rnaSPAdes; VL, Velvet; TA, Trans-ABySS; IT, IDBA-Tran. Mapping rate is defined as percentage of reads aligned (mapped) to the constructed contigs among all reads.

mapping rate, N50 contig size, and percentages of contigs from the BLAST search and complete BUSCO were evaluated. As shown in Table 3 and Fig 3, integration of the multiple assemblies greatly reduced the total number of contigs, even when using mixed raw reads from three transcriptome data. The mapping rate of raw reads to the obtained contig was >85% for most patterns, but less than 70% for other patterns (Assembly 3, 10, and 14). The N50 contig size was around 1200–1300 bp, and there were no major differences between the combination patterns. The hit rates for the BLAST search of the contigs ranged from 23.4% to 30.4%, indicating that assembly integration had not improved much. Rate of complete BUSCO was 92.3–93.5% for all patterns, and notably, the rate of single-copy BUSCO improved to a high value when compared to not merging the assemblies. From the results of the BUSCO assessment, combining assemblies by applying EvidentialGene can greatly reduce redundancy without compromising the completeness of the assemblies. In each evaluation, the total number of contigs was the lowest (129,858 contigs) in Assembly 2 (Trinity and Velvet), and the mapping rate was highest (89.7%) in Assembly 5 (rnaSPAdes and Velvet). In Assembly 4 (Trinity and IDBA-Tran), the rate of the BLAST hit contig was highest (30.4%) and N50 was longest (1,392 bp). The rate of complete BUSCO was highest (93.5%) in Assembly 15 (Trinity, Velvet, and IDBA-Tran). Overall, Assembly 2 (Trinity and Velvet) and Assembly 4 (Trinity and IDBA-Tran) yielded good results; the total number of contigs was decreased while maintaining completeness. In particular, we deemed that Assembly 4 was the best based on the higher mapping rate (89.1%) and rates of BLAST hit contigs and complete BUSCO.

We tested the effect of integrating assemblies using one or two of the three transcriptome data sets to see if a similar trend was observed depending on the number of raw reads (S2 Table). As in the case of mixing three transcriptome data sets, the total number of contigs was

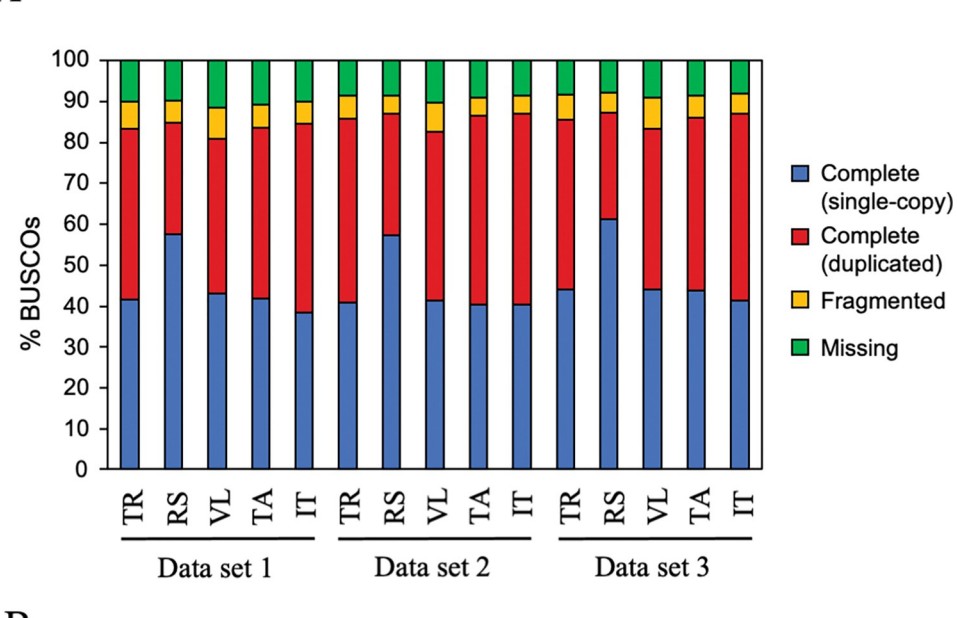

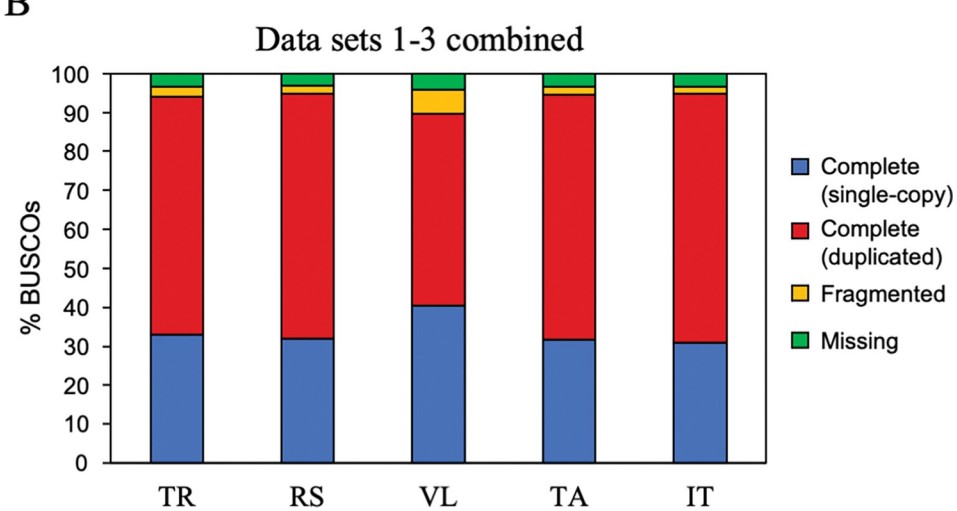

**Fig 1. Cumulative percentage of orthologues identified from the BUSCO search for assemblies shown in Table 1.**
A. BUSCO percentages of assemblies generated by the five assemblers using one transcriptome data set. B. BUSCO
percentages of assemblies obtained by five assemblers using three combined transcriptome data sets. Details for each
transcriptome data and abbreviations for assemblers are shown in Table 1.

reduced, and the percentage of complete and single-copy BUSCO increased by combining out-
put assemblies derived from different assembler. When using one or two transcriptome data,
the mapping rate was high (around 90%) in Assembly 1 (Trinity and rnaSPAdes) and Assem-
bly 5 (rnaSPAdes and Velvet). Assembly 2 (Trinity and Velvet), Assembly 4 (Trinity and
IDBA-Tran) and Assembly 9 (Velvet and IDBA-Tran) yielded a relatively high percentages of
BLAST hit contigs, while Assembly 13 (Trinity, rnaSPAdes and IDBA-Tran) and Assembly 22
(Trinity, rnaSPAdes, Velvet and IDBA-Tran) yielded high percentages of complete BUSCO.

We then assessed the applicability of the proposed pipeline for existing RNA-seq data from
other orchid species, *Phalaenopsis equestris*, and *Apostasia shenzhenica*. For *P. equestris*, the
transcriptome data derived from root, leaf, and flower tissues were combined for this analysis,

**Table 2. Number and combination pattern of assembly using the five assemblers.**

| Assembly | No. of assembly to mix | Trinity | rnaSPAdes | Velvet | Trans-ABySS | IDBA-Tran |
|---|---|---|---|---|---|---|
| 1 | 2 | ● | ● | ○ | ○ | ○ |
| 2 | 2 | ● | ○ | ● | ○ | ○ |
| 3 | 2 | ● | ○ | ○ | ● | ○ |
| 4 | 2 | ● | ○ | ○ | ○ | ● |
| 5 | 2 | ○ | ● | ● | ○ | ○ |
| 6 | 2 | ○ | ● | ○ | ● | ○ |
| 7 | 2 | ○ | ● | ○ | ○ | ● |
| 8 | 2 | ○ | ○ | ● | ● | ○ |
| 9 | 2 | ○ | ○ | ● | ○ | ● |
| 10 | 2 | ○ | ○ | ○ | ● | ● |
| 11 | 3 | ● | ● | ● | ○ | ○ |
| 12 | 3 | ● | ● | ○ | ● | ○ |
| 13 | 3 | ● | ● | ○ | ○ | ● |
| 14 | 3 | ● | ○ | ● | ● | ○ |
| 15 | 3 | ● | ○ | ● | ○ | ● |
| 16 | 3 | ● | ○ | ○ | ● | ● |
| 17 | 3 | ○ | ● | ● | ● | ○ |
| 18 | 3 | ○ | ● | ● | ○ | ● |
| 19 | 3 | ○ | ● | ○ | ● | ● |
| 20 | 3 | ○ | ○ | ● | ● | ● |
| 21 | 4 | ● | ● | ● | ● | ○ |
| 22 | 4 | ● | ● | ● | ○ | ● |
| 23 | 4 | ● | ● | ○ | ● | ● |
| 24 | 4 | ● | ○ | ● | ● | ● |
| 25 | 4 | ○ | ● | ● | ● | ● |
| 26 | 5 | ● | ● | ● | ● | ● |

Black-filled circles indicate that the Assembly shown in the row uses the assembler in the column heading, open circles not.

and those from seed, pollen, and tuber tissues were combined for *A. shenzhenica*. The genome of these two orchids has already been sequenced and the number of protein-coding sequences was estimated [46,47]. In addition to the total number of contigs, mapping rate, N50 contig size and complete BUSCO, we compared the number of predicted genes and the sequence identities for the assemblies obtained with our pipeline and then evaluated whether the completeness of each assembly was valid. The results showed that, when assemblies were integrated, the total number of contigs decreased and the rate of complete and single-copy BUSCO increased (S3 Table). In *P. equestris*, 29,431 genes were predicted as protein-coding genes, and 15,530 of these genes were regarded as a high-confidence gene set [46]. The number of protein-coding genes we predicted by *de novo* assembly from *P. equestris* RNA-seq data ranged from 19,319 to 22,174 when using one assembly and about 16,000 when using multiple assemblies, suggesting that integration of multiple assemblies can be used for more accurate predictions of gene sequences by reducing the redundancy of contigs generated through *de novo* assembly. In *P. equestris*, the mean values of sequence identity were around 98% and the median values were around 99%, and no noticeable difference among assemblies was observed (S3 Table). Similarly for *A. shenzhenica*, integration of assemblies reduced the number of contigs while increasing the percentage of single-copy BUSCO, and successfully constructed accurate assemblies based on the number of predicted genes and sequence identities.

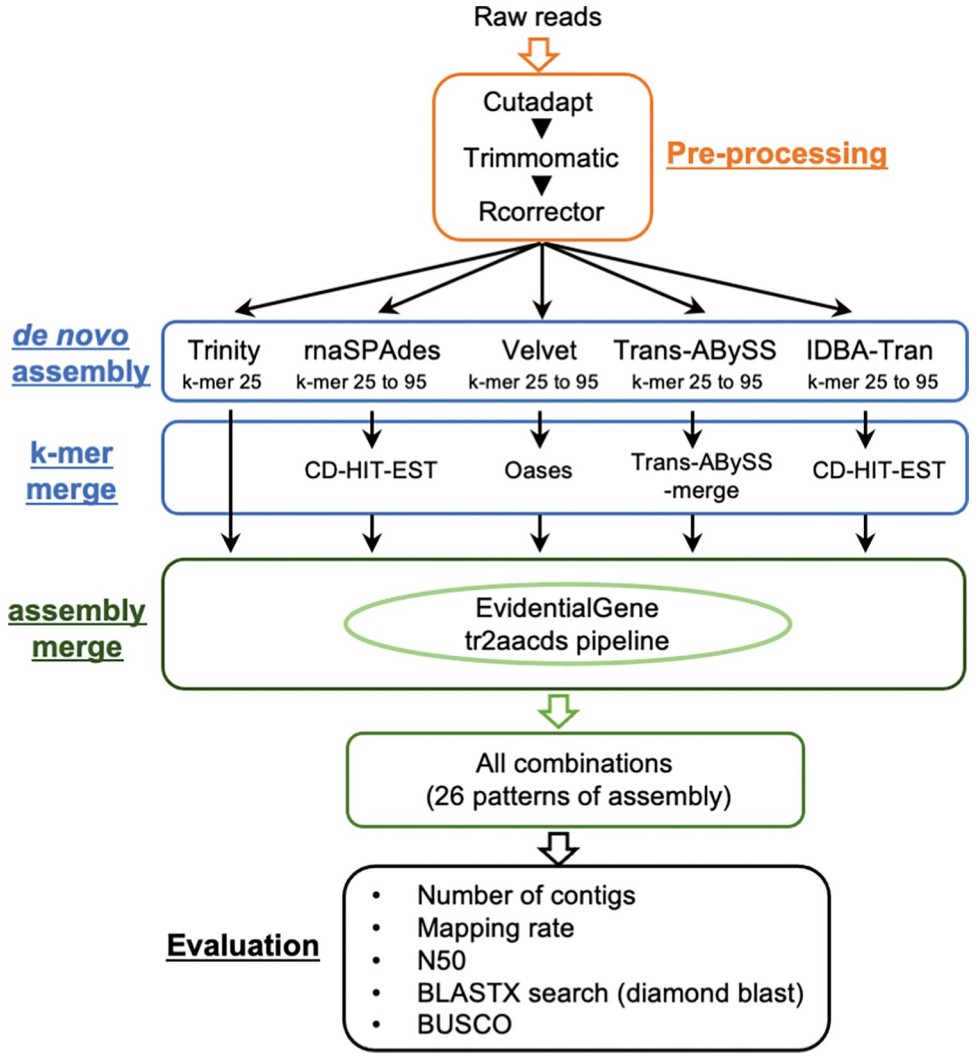

**Fig 2. Flowchart of assembly construction used in study.**

From the results above, we chose Assembly 4 (Trinity and IDBA-Tran) as the best assembly (Fig 4), and then we examined the validity of estimating gene expression using this contig set as a reference. In a mutualistic association such as a rhizobium–legume interaction or AM interaction, nodulin-like genes and common symbiotic genes (CSGs) required for establishing the symbiotic relationship have been proposed [15–17]. We looked for putative nodulin-like genes and CSGs homologs in the CDS region in Assembly 4. All the identified CSG genes (*CYCLOPS*, *NUP85*, *CCaMK*, *CASTOR*, *NUP133*, *POLLUX* and *NENA*) and one nodulin-like gene were annotated to contigs, indicating *C. macranthos* var. *rebunense* also has these genes. We estimated the expression levels of these genes by values of normalized read count (transcripts per million, TPM) in different culture conditions (S2 Fig). Among putative CSGs homologs, *CCaMK*, *CASTOR* and *CYCLOPS* and nodulin-like gene were presumed to highly express in the OM fungus-infected protocorms. Trehalose has been suggested as a fungi-derived carbon source for orchids; in *Dactilorhiza majalis*, the treatment by validamycin A (trehalase inhibitor) inhibited the growth of the protocorms infected with *Ceratobasidium* sp. [8]. In the Assembly 4, six contigs had high sequence similarity with the trehalase genes, and

**Table 3. Evaluation of 26 assemblies constructed by integrating assemblies from different assemblers.**

| Assembly | Total no. of contigs | Mapping rate | N50 | % of BLAST hit contigs |
|---|---|---|---|---|
| 1 | 1,52,649 | 89.3 | 1,336 | 27.0 |
| 2 | 1,29,858 | 89.5 | 1,290 | 28.8 |
| 3 | 1,48,441 | 69.8 | 1,275 | 26.9 |
| 4 | 1,31,656 | 89.1 | 1,392 | 30.4 |
| 5 | 1,59,284 | 89.7 | 1,271 | 26.2 |
| 6 | 1,63,808 | 89.6 | 1,278 | 25.4 |
| 7 | 1,51,750 | 87.9 | 1,336 | 26.9 |
| 8 | 1,48,955 | 76.0 | 1,238 | 26.3 |
| 9 | 1,35,069 | 89.1 | 1,327 | 29.8 |
| 10 | 1,46,162 | 63.7 | 1,324 | 27.6 |
| 11 | 1,65,251 | 88.8 | 1,238 | 25.4 |
| 12 | 1,68,759 | 88.4 | 1,243 | 24.8 |
| 13 | 1,58,652 | 87.3 | 1,289 | 26.0 |
| 14 | 1,59,932 | 67.1 | 1,189 | 25.3 |
| 15 | 1,46,814 | 88.5 | 1,270 | 27.9 |
| 16 | 1,55,400 | 88.7 | 1,270 | 26.4 |
| 17 | 1,74,209 | 89.1 | 1,199 | 24.1 |
| 18 | 1,65,063 | 87.5 | 1,236 | 25.1 |
| 19 | 1,67,508 | 87.3 | 1,250 | 24.7 |
| 20 | 1,58,771 | 89.2 | 1,227 | 25.8 |
| 21 | 1,77,511 | 88.1 | 1,176 | 23.8 |
| 22 | 1,69,210 | 87.0 | 1,207 | 24.7 |
| 23 | 1,71,467 | 86.5 | 1,220 | 24.3 |
| 24 | 1,65,336 | 88.4 | 1,192 | 25.1 |
| 25 | 1,76,761 | 87.1 | 1,179 | 23.6 |
| 26 | 1,79,155 | 86.4 | 1,163 | 23.4 |

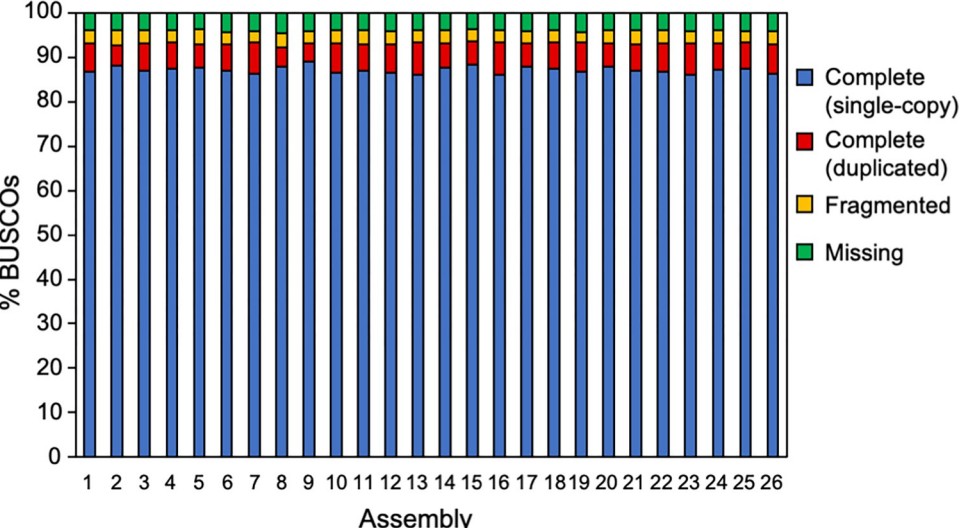

**Fig 3. BUSCO percentages of constructed assembly based on 26 patterns shown in Table 3.**

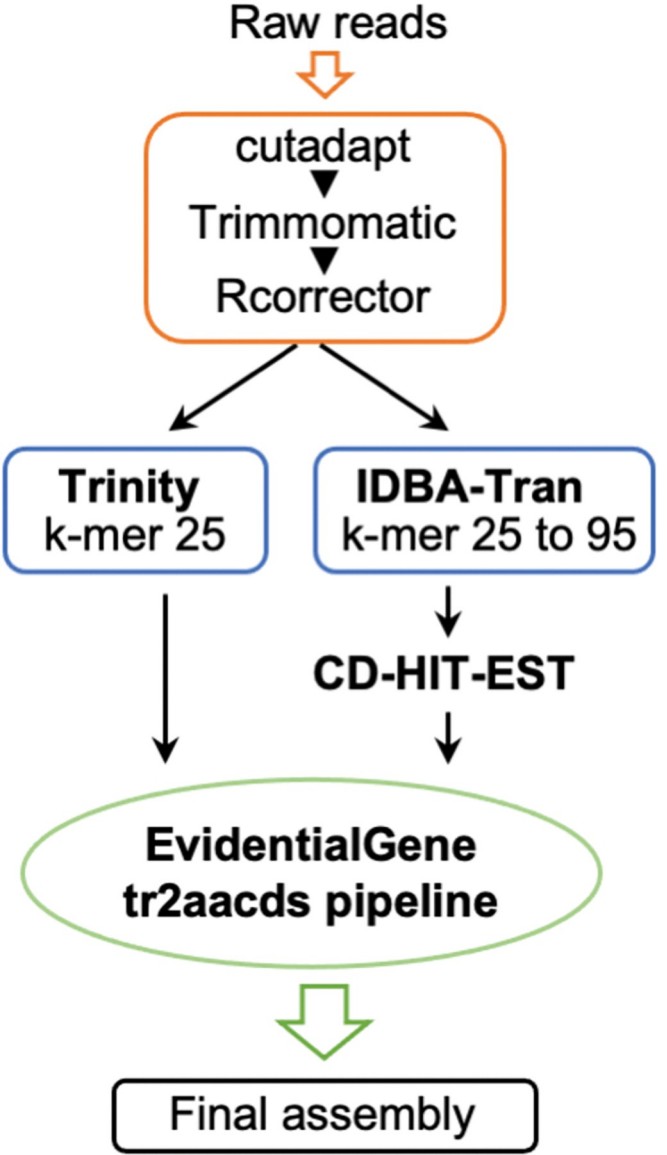

**Fig 4. Optimal pipeline devised in this study for assembly construction using multiple sets of transcriptome data of *Cypripedium*.**

among them, read counts of three putative trehalase genes (evgLocus_101728.p1, evgLocus_593495.p1, and evgLocus_117844.p1) were highly detected in OM fungus-infected protocorms, but not in aseptically grown protocorms (S2 Fig). As carbohydrate-related genes, the *SWEET* (*Sugars Will Eventually be Exported Transporter*) gene family has also been suggested to be involved in OM interactions [11]. In both OM fungus-infected protocorms, read counts on the *SWEET14* gene were highly detected compared to aseptically grown protocorms (S2 Fig). The trend of computational expression levels of genes involved in the OM interactions were consistent with existing reports [11,14], confirming that Assembly 4 contigs can be used to estimate gene expression levels.

To further validate the use of Assembly 4 in downstream analyses of RNA-seq, we conducted DEGs analysis. Because not enough protocorms were available to prepare biological

replicates, the biological coefficient of variation was set as 0.2 according to Chen et al. (2016) [48]. When count data mapped to the contigs of Assembly 4 were compared between protocorms grown with OM fungi and those without fungi, 6123 DEGs and 7968 DEGs were detected in FT061-infected and WO97-infected protocorms, respectively (S3 Fig). Among the DEGs, 3029 DEGs were common to both. To see functional aspects of the DEGs, DEGs were separated into three groups (FT061-WO97-shared group; DEGs common to FT061- and WO97-infected protocorms; FT061-specific group; DEGs specific to FT061-infected protocorms; WO97-specific group; DEGs specific to WO97-infected protocorms) and subjected to a GO enrichment analysis. Significantly enriched for GO terms in biological process for each group are shown in S4 Fig. In FT061-WO97-shared group, GO terms related to biotic stress responses such as "systemic acquired resistance" and "defense response" were enriched; in these GO terms, stress/defense responsive genes encoding pathogenesis-related (PR) protein, cysteine-rich receptor-like kinase or hypersensitive-induced protein were included. In FT061-specific and WO97-specific groups, the GO term "carbohydrate metabolic process" was common, and in this GO term, the acidic endochitinase genes were included; read counts on these genes were highly detected in the both OM fungi-infected protocorms. Although more sample replicates are needed for statistical validation, the detected DEGs could be used to analyze important gene functions and pathways in OM interactions.

## Discussion

Because gene expression differs depending on growth conditions, one way to obtain a *de novo* assembly with more comprehensive gene information is to combine transcriptome data obtained under multiple growth conditions and perform a *de novo* assembly. In this study, by mixing raw reads from three transcriptome data sets that were obtained from orchid protocorms cultured in different conditions, the percentage of complete BUSCO improved but that of BLAST hit contigs decreased probably due to greatly increased total number of contigs (Table 1; Fig 1). However, by integrating assemblies with the EvidentialGene tr2aacds pipeline (Fig 2), the total number of contigs decreased without diminishing the complete BUSCO percentage (Table 3; Fig 3). These results suggest that to obtain a complete and less redundant assembly, it is useful to mix raw reads in advance, process for *de novo* assembly using multiple assemblers, and then integrate them using the EvidentialGene tr2aacds pipeline. The effectiveness of the EvidentialGene tr2aacds pipeline was also shown by Nakasugi et al. (2014) and Chen et al. (2015) [27,49]. For RNA-seq study of species for which genomic sequences are not available such as *Cypripedium*, the pipeline proposed here would be useful.

The pipeline that combines the assemblies from Trinity and IDBA-Tran was the best in this study because these combinations had the fewest contigs and yielded a relatively high percentages of BUSCO and BLAST hit contigs. In comparison of each of the five assemblies from mixed reads without integration by EvidentialGene, Trinity had the fewest contigs, followed by IDBA-Tran, and these two assemblers yielded assemblies with a higher percentage of BLAST hit contigs. Therefore, the better result obtained from combining the use of Trinity and IDBA-Tran is thought to be a continuation of the characterization of the outputs from these two assemblers. There was also a trend that as the number of input assemblies into EvidentialGene increased, the total number of contigs increased, but the percentage of complete BUSCO did not change (Table 3; Fig 3). From these results, to obtain a good contig set, choosing less redundant and more complete assemblies for input and putting such assemblies into the EvidentialGene tr2aacds pipeline are appropriate. Assemblies obtained from Velvet sometimes yielded contig sets with lower mapping rates and percentages of complete BUSCO compared to the other four assemblers (Fig 1, Tables 1 and S2). This may be because Velvet is an

assembler developed for genome assembly and has high performance for resolving repeat sequences; Velvet is not supposed for the use in *de novo* assembly of transcriptome data. The other four assemblers (Trinity, rnaSPAdes, Trans-ABySS and IDBA-Tran) are all designed for *de novo* transcriptome assembly, and these assemblers did not construct assemblies with low mapping rates or percentages of complete BUSCO as observed in Velvet. There is an example where Velvet are applied to transcriptome assembly [27], and in our analysis, Velvet could construct good contigs in combination with other assemblers, but it is better to consider that the use of Velvet may be unstable for *de novo* transcriptome assembly in some cases.

When using RNA-seq reads from *P. equestris* and *A. shenzhenica* (S3 Table), we found that the use of the EvidentialGene tr2aacds pipeline reduced the total number of contigs without reducing the complete BUSCO rate, but also reduced the number of genes detected. The EvidentialGene tr2aacds pipeline classifies the input contig set into "Main", "Alternate" and "Drop" based on the sequence similarity using fastanrdb (exonerate; [50]), CD-HIT-EST and blastn [49]. "Main" comprises transcripts expected to have a unique CDS, "Alternate" comprises transcripts that are considered as isoforms, and "Drop" comprises contigs determined to be inappropriate as final outputs. In this study, we used only the "Main" transcripts (i.e., contigs classified as "Alternate" were lost), thus likely reducing the number of genes detected. The application of the pipeline used in this study is useful for analyses such as expression analyses when too many contigs would cause problems with downstream analysis, but in other cases (e.g., detection of mutations or isoforms), desired information may be lost during assembly integration. If variant sequences are desired, using the "Main" and "Alternate" transcripts as the final output from the EvidentialGene tr2aacds pipeline may be a good approach.

In order to evaluate the validity of the contig set constructed in this study and to get any insights about genes important for the interaction between orchids and OM fungi, we estimated gene expression levels in our protocorm samples using Assembly 4 contig set as a reference. Nodulin-like genes, CSGs and *SWEET* genes have been shown to have roles in mutualistic plant–fungus interactions [51–56], and recently, the trehalase gene has been suggested to be involved in mycoheterotrophy in orchids [8,9]. When estimated expression levels of these genes, they were expected to be upregulated in the protocorms grown with OM fungi compared to those in the aseptic condition (S2 Fig). These results were in accordance with those for orchid–fungus interaction in previous reports [11,14], suggesting the validity of the contig set obtained from our constructed pipeline. Using Assembly 4 contig set, we also attempted DEGs and GO enrichment analyses and found that GO terms "systemic acquired resistance" and "defense response" were enriched for the shared DEGs in OM fungi-infected conditions. Because we were unable to prepare sufficient replicates necessary to apply the statistical evaluation this time, we cannot debate biological meaning about the results from DEG and GO analyses, but we presume that orchids deploy a more aggressive defense response by using symbiotic fungus-specific antifungal compounds [10]. There are still many unknowns in the OM interaction compared to AM interaction. To determine whether the OM relationship is benign, beneficial or competitive, or somewhere in between, further research is necessary to better understand this unique interaction between orchids and fungi.

## Supporting information

**S1 Fig. Representative images of protocorms used for RNA-seq.** A. Protocorms formed by infection with OM fungi (strain W097 or FT061). Note that W097-infected protocorms stopped growing, but FT061-infected protocorms continued to grow and underwent initial root differentiation (arrows). B. Protocorms obtained in aseptic conditions.
(TIF)

**S2 Fig. Comparison of estimated expression levels of putative genes involved in mutualistic association with microbes in OM fungus-infected protocorms or aseptically grown protocorms.** Transcriptome abundance was estimated by mapping the pre-processed reads to the reference contig set using Salmon vl .7.0 and shown by a value of transcripts per million (TPM).
(TIF)

**S3 Fig. Differentially expressed gene (DEG) analysis of Cypripedium transcriptome data.** DEGs between OM fungus- infected protocorms vs aseptically grown protocorms were detected using edgeR with a false discovery rate (FDR) <0.05. Because not enough protocorms were available to prepare biological replicates, the biological coefficient of variation was set as 0.2 according to Chen et al. (FlOOORresearch. 2016; 5: 1438). In a graph, total number of detected DEGs is shown above a bar.
(TIF)

**S4 Fig. Significantly enriched Gene Ontology (GO) terms related to "Biological process" for DEGs detected in OM fungus-infected protocorms.** GO enrichment was analyzed by using Fisher's exact test with FDR <0.05 to test whether the proportion of genes annotated with a GO term among the DEGs was significantly greater than that among the entire contigs. GO terms were annotated using the contig set and InterProScan vl.8.0. FT061-WO97-shared group, DEGs common to W097- and FT061- infected protocorms; FT061-specific group, DEGs specific to FT061-infected protocorms; W097-specific group, DEGs specific to W097-infected protocorms.
(TIF)

**S1 Table. Annotation of the contigs from the DIAMOND blastx search.**
(XLSX)

**S2 Table. Evaluation of 26 assemblies constructed by integrating assemblies using one or two transcriptome data sets.**
(XLSX)

**S3 Table. Evaluation of 26 assemblies constructed by integrating assemblies using transcriptome data from *Phalaenopsis equestris* and *Apostasia shenzhenica*.**
(XLSX)

## Acknowledgments

Computational analysis was partially performed on the NIG supercomputer at the Research Organization of Information and Systems (ROIS), National Institute of Genetics. We would like to thank Dr. Tetsuo Takano and Dr. Daisuke Tsugama (The University of Tokyo, Japan) for allowing us to use their computer.

## Author Contributions

**Conceptualization:** Kaien Fujino, Hanako Shimura.

**Data curation:** Kota Kambara.

**Funding acquisition:** Hanako Shimura.

**Investigation:** Kota Kambara.

**Methodology:** Kota Kambara, Hanako Shimura.

**Resources:** Hanako Shimura.

**Supervision:** Kaien Fujino.

**Validation:** Kota Kambara, Kaien Fujino, Hanako Shimura.

**Visualization:** Kota Kambara.

**Writing – original draft:** Kota Kambara, Hanako Shimura.

**Writing – review & editing:** Kaien Fujino, Hanako Shimura.

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
