## [Decision Letter · Decision Letter 0]

6 Apr 2023

PONE-D-23-08120Construction of a de novo assembly pipeline using multiple transcriptome data sets from Cypripedium macranthos (Orchidaceae)PLOS ONE

Dear Dr. Shimura,

Thank you for submitting your manuscript to PLOS ONE. After careful consideration, we feel that it has merit but does not fully meet PLOS ONE’s publication criteria as it currently stands. Therefore, we invite you to submit a revised version of the manuscript that addresses the points raised during the review process.

We look forward to receiving your revised manuscript.

Kind regards,

Matthew Cserhati, Ph.D

Academic Editor

PLOS ONE

Journal Requirements:

Additional Editor Comments:

Besides the comments made by the reviewer, I also have the following observations regarding th manuscript:

1. Several typos occur in the text; the English could use a little bit of improvement.

2. Please define basic terms, such as protocorm.

3. Lines 21-23: “The family Orchidaceae comprises the most species of any monocotyledonous family and has interesting characteristics such as seed germination induced by mycorrhizal fungi and flower morphology that co-evolved with pollinators.” Is incorrect, it is more accurate to say that the fungi and the flower morphology co-adapted with pollinators. We are talking about already existing structures that underwent adaptation.

4. Line 190: table 1: incomplete value: 1048,92

5. Why did the VL assembler have such a low mapping rate in data sets 1 and 3? Does this nit skew the results? Shouldn’t this assembler be excluded? Would the results change that much?

Reviewers' comments:

Reviewer's Responses to Questions

**Comments to the Author**

1. Is the manuscript technically sound, and do the data support the conclusions?

Reviewer #1: Partly

2. Has the statistical analysis been performed appropriately and rigorously? 

Reviewer #1: Yes

3. Have the authors made all data underlying the findings in their manuscript fully available?

Reviewer #1: No

4. Is the manuscript presented in an intelligible fashion and written in standard English?

Reviewer #1: Yes

5. Review Comments to the Author

Reviewer #1: In this paper, Kambara et al. reported the establishment of a de novo assembly pipeline for transcriptome data from the wild orchid Cypripedium. Using the contig set obtained by the established method, analyses of differentially expressed genes (DEGs) regarding mycorrhizal formation and maintenance were conducted. I agree with the authors’ idea that the de novo assembly pipeline established in this paper is effective and useful for the species for which genome sequences are not available such as Cypripedium to conduct RNA-seq. However, I have serious concerns about the statements of results and discussion based on the afterward transcriptome analysis because no biological replicates of samples were available.

For example, the authors showed TPM value based on one sample data, but at least three biological replicates are generally demanded to demonstrate the gene expression levels (e.g., qRT-PCR). Generally, DEGs should be obtained based on reliable statistical analysis. Therefore, I am regret to say that GO enrichment analysis of the DEGs obtained from no biological replicate work has no biological meaning. I understand that the authors would like to validate the established pipeline by the afterward transcriptome analysis. However, I think the statement should be more restricted, considering the results obtained from insufficient replicates. Hence, I would like to propose that Fig. 4-6 are moved to supplemental information, and the statement regarding this part should focus on the validation. In particular, the discussion part (Line 389-421) is overdiscussed. Hence, I recommend that the contents of this part should be reconsidered and shortened as far as possible.

In addition, I noticed that the abstract does not contain a conclusion at the end. Likewise, which content is the point the authors’ would like to claim to readers in this paper is not clear in the discussion part. Hence, please reconsider these contents and compositions.

On the other hand, I think the validation of the assemblies using the data of genome-sequenced orchids is not enough because the authors compared only the number of predicted genes. The authors suggested the accuracy of assembling only based on the number of predicted genes, but I think that the accuracy should be shown based on the sequence identity. Thus, please add such analyzed data. I think the top five list summarized in Table 4 is also not required. In addition, the assembled sequence data obtained in this study is not available, and only the unique gene IDs were shown in the manuscript. Considering that other researchers may verify or use the results obtained in this study, I strongly recommend that the representative assembled data should be deposited in public database database/website or submitted as supplemental data. I hope that the authors will revise the manuscript according to my comments.

Minor comments:

Line 45: ‘orchid mycorrhiza fungi’ should be ‘orchid mycorrhizal fungi’.

Line 64: ‘et al’ should be ‘et al.’ (same afterward)

Line 72: Words at the beginning of a sentence are not abbreviated. (same afterward)

Line 73: Considering the content of previous sentence, ‘However’ is not a suitable word for connecting.

Line 108: The expression of ‘12-14 weeks’ is not suitable to show the timing to harvest each sample because two weeks range is so different. Please reconsider and show it for each sample.

Line 109: Please state the details about the strains WO97 and FT061. The current statement is not enough to understand what the authors used.

Line 226: What is ‘mapping rate’? I could not get its meaning. (same afterward)

6. PLOS authors have the option to publish the peer review history of their article (what does this mean?). If published, this will include your full peer review and any attached files.

Reviewer #1: No

---

## [Author Response · Author response to Decision Letter 0]

27 Apr 2023

To Editor,

Thank you for your comments. We addressed each of the your comments in a file of “Response to Reviewers”

To Reviewer 1,

Thank you for your comments. We agree with your opinion that we should not state discussions about the results for gene expression in the absence of proper biological replicates. We addressed each of the your comments in a file of “Response to Reviewers”

---

## [Decision Letter · Decision Letter 1]

24 May 2023

Construction of a de novo assembly pipeline using multiple transcriptome data sets from Cypripedium macranthos (Orchidaceae)

PONE-D-23-08120R1

Dear Dr. Shimura,

We’re pleased to inform you that your manuscript has been judged scientifically suitable for publication and will be formally accepted for publication once it meets all outstanding technical requirements.

Kind regards,

Matthew Cserhati, Ph.D

Academic Editor

PLOS ONE

Additional Editor Comments (optional):

Reviewers' comments:

Reviewer's Responses to Questions

**Comments to the Author**

1. If the authors have adequately addressed your comments raised in a previous round of review and you feel that this manuscript is now acceptable for publication, you may indicate that here to bypass the “Comments to the Author” section, enter your conflict of interest statement in the “Confidential to Editor” section, and submit your "Accept" recommendation.

Reviewer #1: All comments have been addressed

2. Is the manuscript technically sound, and do the data support the conclusions?

Reviewer #1: Yes

3. Has the statistical analysis been performed appropriately and rigorously? 

Reviewer #1: Yes

4. Have the authors made all data underlying the findings in their manuscript fully available?

Reviewer #1: Yes

5. Is the manuscript presented in an intelligible fashion and written in standard English?

Reviewer #1: Yes

6. Review Comments to the Author

Reviewer #1: (No Response)

7. PLOS authors have the option to publish the peer review history of their article (what does this mean?). If published, this will include your full peer review and any attached files.

Reviewer #1: No

---

## [Editor Report · Acceptance letter]

29 May 2023

PONE-D-23-08120R1 

Construction of a *de novo* assembly pipeline using multiple transcriptome data sets from *Cypripedium macranthos* (Orchidaceae) 

Dear Dr. Shimura:

I'm pleased to inform you that your manuscript has been deemed suitable for publication in PLOS ONE. Congratulations! Your manuscript is now with our production department. 

Kind regards, 

on behalf of

Dr. Matthew Cserhati 

Academic Editor

PLOS ONE